# Towards Open-Vocabulary Semantic Segmentation Without Semantic Labels

**Heeseong Shin**[1]     **Chaehyun Kim**[1]     **Sunghwan Hong**[2]     **Seokju Cho**[1]

**Anurag Arnab**[†,3]     **Paul Hongsuck Seo**[†,2]     **Seungryong Kim**[†,1]

[1]KAIST     [2]Korea University     [3]Google Research

{hsshin98, kchyun, seokju.cho, seungryong.kim}@kaist.ac.kr[1]
{sung_hwan, phseo}@korea.ac.kr[2]     aarnab@google.com[3]

## Abstract

Large-scale vision-language models like CLIP have demonstrated impressive open-vocabulary capabilities for image-level tasks, excelling in recognizing *what* objects are present. However, they struggle with pixel-level recognition tasks like semantic segmentation, which additionally require understanding *where* the objects are located. In this work, we propose a novel method, **PixelCLIP**, to adapt the CLIP image encoder for pixel-level understanding by guiding the model on *where*, which is achieved using unlabeled images and masks generated from vision foundation models such as SAM and DINO. To address the challenges of leveraging masks without semantic labels, we devise an online clustering algorithm using learnable class names to acquire general semantic concepts. PixelCLIP shows significant performance improvements over CLIP and competitive results compared to caption-supervised methods in open-vocabulary semantic segmentation. Project page is available at https://cvlab-kaist.github.io/PixelCLIP

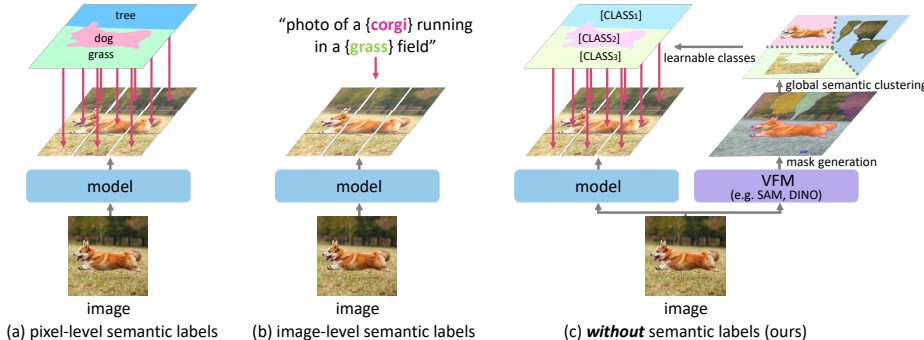

Figure 1: **Illustration of different approaches for open-vocabulary semantic segmentation.** In contrast to existing methods utilizing (a) pixel-level semantic labels [1, 2, 3, 4, 5, 6] or (b) image-level semantic labels [7, 8, 9, 10, 11, 12], we leverage unlabeled masks as supervision, which can be freely generated from vision foundation models such as SAM [13] and DINO [14].

## 1 Introduction

Semantic segmentation is a fundamental task in computer vision where the goal is to identify class labels for each pixel within the given image. However, segmentation datasets often require extensive human effort to obtain densely-annotated semantic labels, limiting their scalability. In this regard, recent advances in large-scale pre-trained vision-language models, *e.g.* CLIP [15] and ALIGN [16],

---

[†]Corresponding authors

38th Conference on Neural Information Processing Systems (NeurIPS 2024).

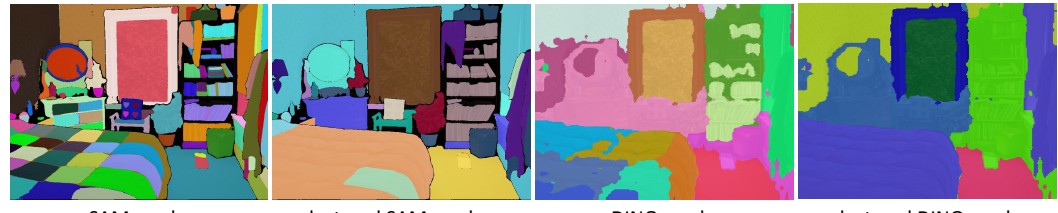

| SAM masks | clustered SAM masks | DINO masks | clustered DINO masks |

Figure 2: **Visualization of masks from vision foundation models.** We visualize the masks generated by SAM [13] and by clustering image features from DINO [14]. Although such models can freely generate fine-grained masks, the resulting masks can be too small or incomplete to have semantic meaning. To address this over-segmentation issue, we employ online clustering [18] of the masks into semantically meaningful groups defined globally for given images.

have facilitated open-vocabulary semantic segmentation [1, 3, 2, 17, 4, 6], which aims to generalize semantic segmentation into unbounded range of classes. Despite showing remarkable generalization capabilities, they still require pixel-level semantic labels for leveraging the image-level pre-trained vision-language models for semantic segmentation.

Recently, several studies [11, 12, 7, 8] have pioneered open-vocabulary semantic segmentation without densely-annotated semantic labels. These studies often utilize image-level semantic labels, such as image captions, to enhance the pre-trained vision-language models like CLIP for semantic segmentation. However, image captions typically provide information about *what* is in the image, but without *where* it is. Since CLIP is already effective in recognizing *what* the objects are, this causes models to only implicitly learn object locations, leading to sub-optimal performance or requiring millions of image-caption pairs to compensate for this weak supervision [8, 7]. Instead, we focus on informing CLIP about *where* objects are located to address the missing information.

In this study, we propose a novel approach to achieve open-vocabulary semantic segmentation *without* leveraging semantic labels, but through guiding the pre-trained vision-language models, such as CLIP, on *where* to look. We leverage recent vision foundation models (VFMs), such as DINO [14] and SAM [13], to partition images into fine-grained regions to indicate *where* to look. Consequently, we explore methods to effectively leverage these masks for fine-tuning the image encoder of CLIP.

In contrast to existing works that leverage semantic labels [6, 19, 7], we do not have any captions or class names that can be fed to the text encoder of CLIP. To leverage its knowledge, we devise a method that employs prompt learning [20, 21] on the text encoder of CLIP to construct *learnable classes*. Setting the learnable classes as a centroid, we propose applying the online clustering algorithm [18, 22] along the given masks to gather them into semantically meaningful groups, as shown in Fig. 2. We keep these learnable classes global across the entire images, which guides the learnable classes to contain the general semantic concepts. Despite the absence of semantic labels, our method is able to jointly leverage the image encoder and text encoder of CLIP during training, successfully achieving dense open-vocabulary recognition.

Our framework, called PixelCLIP, achieves significant improvements to CLIP, on average of +16.2 mIoU in open-vocabulary semantic segmentation. Moreover, despite not using any semantic labels, PixelCLIP shows competitive performance in comparison to image-level supervised methods using captions [7, 9, 10], demonstrating the effectiveness of unlabeled masks for. We further show the effectiveness of PixelCLIP for classifying masks from various open-vocabulary segmentation models, which can be simply done by replacing the CLIP within existing methods. We also provide extensive ablation studies to validate our choices, with a detailed analysis of our method.

We summarize our contribution as follows:

- We propose a novel formulation of learning from images *without* semantic label for open-vocabulary semantic segmentation by leveraging masks generated from DINO and SAM to fine-tune vision-language models.
- We propose to globally cluster semantically similar masks by employing an online clustering algorithm, while learning class prompts for representing semantic clusters.
- We demonstrate significant gains in open-vocabulary semantic segmentation, even surpassing methods leveraging image-level semantic labels, and provide thorough ablation studies with analysis to validate our framework.

## 2 Related Work

### 2.1 Open-vocabulary semantic segmentation

Open-vocabulary semantic segmentation [2, 23] aims to label each pixel within an image into an unbounded range of classes. In this regard, recent works [1, 17, 2, 6, 24] aim to generalize to classes unseen during training through leveraging pre-trained vision-language models, such as CLIP [15]. Despite their remarkable performance, they leverage per-pixel semantic labels during their training, which requires expensive cost to annotate. Instead, we focus on the weakly-supervised setup, where the goal is to zero-shot transfer to segmentation task *without* densely-annotated class labels [11, 12, 25, 5, 26, 7, 27], utilizing image-level labels as supervision or even no labels at all.

In this regard, recent studies [11, 12, 25, 5] leverage image caption as supervision. GroupViT [11] and ViL-Seg [12] are pioneering works for identifying groups or clusters emerging from captions. Along with the advance of vision-language models, SegCLIP [9] and TCL [7] leverage pre-trained CLIP and learn additional decoder modules to learn dense vision-language alignment. PACL [8] learns additional embedding layers to enhance the patch-level alignment in vision-language models and SAM-CLIP [10] attempts to merge SAM [13] and CLIP [15] into a unified model by additionally leveraging unlabeled mask data from SAM. Apart from these approaches, we avoid employing *any* semantic labels [26, 28], but leverage vision foundation models to obtain masks as a source for supervision to fine-tune the CLIP image encoder for achieving open-vocabulary semantic segmentation.

### 2.2 Fine-tuning vision-language models for dense prediction

Recent large-scale pre-trained vision-language models have shown its effectiveness for jointly understanding images and language [29, 15, 16]. Notably, CLIP [15], trained with web-scale image-caption pairs, has been widely popularized for transferring its open-vocabulary recognition capabilities to various downstream tasks [26, 30, 31, 32]. However, despite its success in image-level tasks like image classification, CLIP tends to struggle in dense prediction tasks [17, 26, 6], such as object detection and semantic segmentation. This originates from CLIP being trained from image-level supervision being captions, hence exhibits bias towards the global image rather than fine-grained regions within the image [17]. While non-learnable approaches, such as MaskCLIP [26] show improvements by slightly modifying the architecture, CLIP still shows limited capabilities in dense predictions in comparison to its global understanding.

To address this, OWL-ViT [33] directly fine-tunes pre-trained vision and text encoders to downstream open-vocabulary detection task, and CAT-Seg [6] introduces a cost aggregation scheme for fine-tuning the encoders of CLIP for semantic segmentation. Alternatively, ZegCLIP [19] and Xu et al. [3] implement prompt tuning [21, 20] for tuning the image and text encoders of CLIP. Instead of fine-tuning the full model, they learn prompt tokens that serve as a global prefix for the encoders of CLIP. While such methods show remarkable results from fine-tuning the encoders of CLIP for dense downstream tasks, they require densely annotated detection and segmentation data for training.

### 2.3 Vision foundation models

With the advent of large-scale learning enabled by scalable vision backbone architectures [34, 35] and vast amounts of data, diverse vision foundation models are emerging in the field of computer vision. In this regard, self-supervised methods [36, 37, 38, 39] have demonstrated the effectiveness of its rich visual representations for various downstream tasks. Especially, DINO [14] exerted strengths in fine-grained semantic recognition [40, 41], making it highly effective for object detection and image segmentation. Moreover, DINO features have also been demonstrated for yielding fine-grained masks within the image through applying the $k$-means clustering with its features [42, 43, 44].

On the other hand, the segment anything model (SAM) [13] has demonstrated its capability for generating fine-grained, high-quality segmentation masks for any object in an image. Through its self-annotation pipeline, SAM has collected an unprecedented amount of mask annotation for achieving its capabilities. While we can freely leverage SAM to obtain detailed masks in any given image, we mainly utilize the pre-computed masks within the collected dataset, SA-1B. Both DINO and SAM, however, yield unlabeled masks without semantic labels as both models are also trained without semantic labels, presenting a challenge for leveraging their masks for achieving dense vision-language recognition.

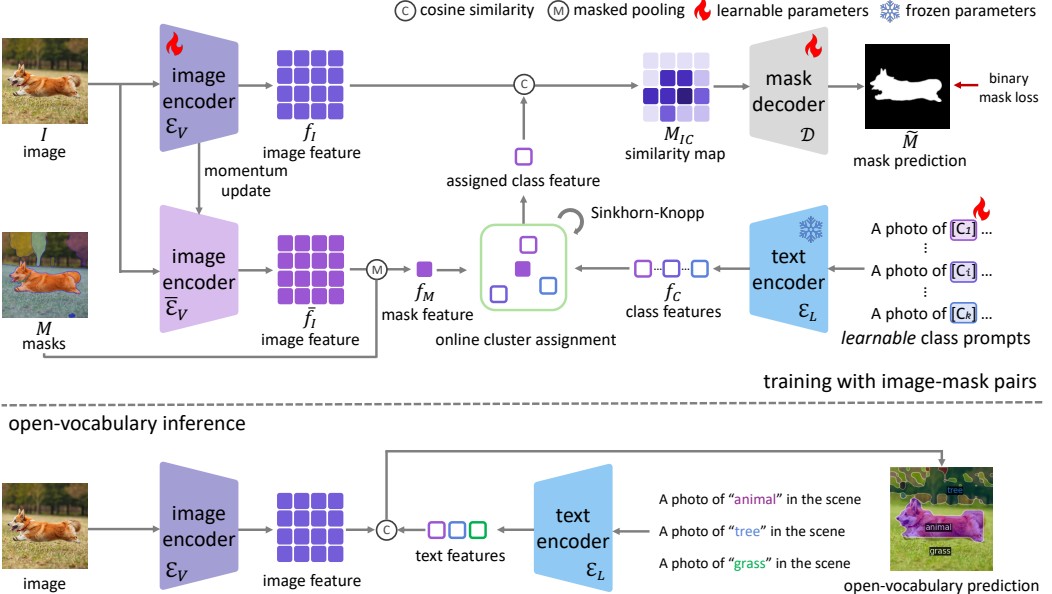

Figure 3: **Illustration of our overall framework.** We provide illustration of PixelCLIP, utilizing unlabeled images and masks for fine-tuning the image encoder of CLIP, enabling open-vocabulary semantic segmentation. We note that the momentum image encoder and the mask decoder are only leveraged during training, and inference is only done with image and text encoders of CLIP.

## 3 Methodology

In this section, we first establish our problem formulation of learning dense vision-language alignment from images paired with masks, generated from vision foundation models. Next, we discuss the challenges of leveraging masks as supervision for fine-tuning the image encoder of CLIP and finally, present our methodology of semantic clustering of masks to address the challenges.

### 3.1 Preliminaries

Given an input image $I \in \mathbb{R}^{H \times W \times 3}$, open-vocabulary semantic segmentation [6, 7] aims to label each pixel within an image with classes given in free-form text. As a training signal, semantic labels offer a set of $S$ textual descriptions for a semantic class $T = \{T_i\}_{i=1}^S$ related to $I$. This can be directly utilized with the CLIP text encoder $\mathcal{E}_L(\cdot)$ to obtain text features $f_T = \mathcal{E}_L(T) \in \mathbb{R}^{S \times d}$, where $d$ is the hidden dimension. Dense image features $f_I = \mathcal{E}_V(I) \in \mathbb{R}^{h \times w \times d}$, where $h \times w$ is the output feature resolution, are then extracted. We finally obtain dense image-text similarity map $M_{IT} \in \mathbb{R}^{h \times w \times S}$:

$$M_{IT}(x, y, n) = \frac{f_I(x, y) \cdot f_T(n)}{\|f_I(x, y)\|\|f_T(n)\|}. \tag{1}$$

This can be interpreted as soft binary masks predicted from image and text features of CLIP, and be supervised with binary mask loss $\mathcal{L}_{\text{mask}}$ in a pixel-level manner to fine-tune CLIP [6].

### 3.2 Integrating masks into CLIP features

In this work, we do not have any access to $T$, but are only given unlabeled masks $M \in \mathbb{R}^{H \times W \times N}$, where $N$ denotes the number of masks for the given image $I$. Hence, we devise methods to predict masks by incorporating $M$ into CLIP features. We aim to fine-tune the CLIP image encoder $\mathcal{E}_V(\cdot)$ through leveraging unlabeled masks $M$ as supervision. Since $M$ is generated from vision foundation models, *e.g.* DINO or SAM, this presents us with the challenge of not having any semantic labels.

In order to integrate masks into CLIP, a straightforward approach would be employing the masks $M$ with the CLIP image feature map $f_I$ to obtain per-mask CLIP features. While there could be various methods to extract regional CLIP features [26, 45, 5], we apply mask pooling over $f_I$ to obtain mask

pooled features $f_M = \text{MaskPool}(f_I, M) \in \mathbb{R}^{N \times d}$. Consequently, we can leverage $f_M$ to obtain image-*mask* similarity map denoted $M_{IM} \in \mathbb{R}^{h \times w \times N}$:

$$M_{IM}(x, y, n) = \frac{f_I(x, y) \cdot f_M(n)}{\|f_I(x, y)\| \|f_M(n)\|}. \tag{2}$$

This allows us to supervise the model with a binary mask loss $\mathcal{L}_{\text{mask}}$ for fine-tuning CLIP with given image $I$ and unlabeled masks $M$. In practice, since $M_{IM}$ has the same resolution as the feature map from the CLIP image encoder $f_I$, we employ a light-weight decoder $\mathcal{D}$ to mitigate the resolution gap between $M_{IM}$ and $M$, as shown in Fig. 3. This can be written as $\mathcal{D} : \mathbb{R}^{h \times w} \to \mathbb{R}^{h' \times w'}$, where $h' \times w'$ is resolution for the upsampled mask. Therefore, the output of the model can be updated as $\tilde{M} = \mathcal{D}(M)$.

## 3.3 Semantic clustering of masks

Upon using mask pooled CLIP image features $f_M$ to predict $M_{IM}$, however, we find the masks generated from DINO and SAM to often over-segment the image, resulting in too small or incomplete masks as seen in Fig. 2. This would require CLIP to forcefully discriminate regions that are semantically similar, impeding the training process.

In this regard, we propose to group semantically similar masks into *clusters* and predict based on the clusters rather than individual masks. Moreover, we aim to define this cluster *globally*, which is shared across the entire training process rather than for each image or iteration. This would be analogous to constructing pixel-level semantic labels, where a fixed set of classes defined over the dataset is equivalent to each cluster. However, the difference is that there is no pre-defined set of classes that we can define the clusters with. While we could heuristically pre-define such classes, we describe our learnable method for globally clustering masks into semantically meaningful groups.

**Online clustering via learnable class prompts.** To globally cluster masks into semantic categories, we propose representing these clusters using CLIP *text* features as centroids for clustering mask features. Given that the CLIP text encoder is trained with a broad understanding of natural language semantics, we expect these clusters to capture meaningful semantics by leveraging its comprehensive pre-trained knowledge. In this regard, we take a learnable approach, where each cluster is defined by class-specific learnable prompts fed into the CLIP text encoder. Unlike existing prompt learning methods, which typically focus on learning a task-specific prefix [20, 21, 3], we aim to learn prompt tokens that represent each class. For instance, in the sentence ''A photo of an object'', traditional prompting methods would learn the tokens for the ''A photo of a'' prefix, whereas our method focuses on learning the token for the ''object.''

Specifically, given the number of clusters $k$, we can define prompt tokens as $C \in \mathbb{R}^{k \times l \times d_e}$, where $l$ is the token length of the prompt and $d_e$ is the dimension of the token embeddings. From this, we can utilize the CLIP text encoder $\mathcal{E}_L(\cdot)$ to obtain a set of *class* features $f_C = \mathcal{E}_L(P^*, C) \in \mathbb{R}^{k \times d}$ in the form of CLIP text features, where $P^*$ is a fixed template for the CLIP text encoder, such as "A photo of a {} in the scene." While we could assign each mask $f_M$ with $f_C$ in a winner-takes-all manner, we desire the classes to encode general semantics across all images. Therefore, we assume that we can equally divide $m$ masks within a minibatch [18, 14], into $k$ clusters given a sufficient amount of masks.

Consequently, we aim to find an assignment $Q \in \mathbb{R}_+^{k \times m}$ based on the image-text similarity between the mask pooled features $f_M$ and the class text features, which can be defined as:

$$\max_{Q \in \mathcal{Q}} \text{Tr}(Q^\top F_M^\top f_C) + \varepsilon H(Q), \quad \text{s.t.} \quad Q \in \mathbb{R}_+^{k \times m}, \quad Q^\top \mathbb{1}_k = \frac{1}{m} \mathbb{1}_m, \quad Q \mathbb{1}_m = \frac{1}{k} \mathbb{1}_k, \tag{3}$$

where $F_M$ is the set of all $m$ mask features $f_M$ within the minibatch, and $\mathbb{1}_k$ denotes the $k$-dimensional vector of ones. $H$ is the entropy function, $H(Q) = -\sum_{ij} Q_{ij} \log Q_{ij}$ with $\varepsilon$ as a hyperparameter. The solution $Q$ from Eq. 3 is an assignment matrix defining which of the $k$ clusters each $m$ mask should belong to, hence $\varepsilon$ determines the smoothness of this mapping Q by scaling the entropy regularization from $H$. The equipartition constraint, $Q^\top \mathbb{1}_k = \frac{1}{m} \mathbb{1}_m, Q \mathbb{1}_m = \frac{1}{k} \mathbb{1}_k$ encourages the class features $f_C$ to be selected at least $m/k$ times on average, allowing to learn general concepts represented by the masks within the dataset. In practice, with the soft assignment relaxation [46], $Q$

can be solved as follows:

$$Q = \operatorname{diag}(u) \exp\left(\frac{F_M^\top f_C}{\varepsilon}\right) \operatorname{diag}(v), \tag{4}$$

where $u \in \mathbb{R}^k$, $v \in \mathbb{R}^m$ denote renormalization vectors, which can be efficiently computed by Sinkhorn-Knopp algorithm [46].

Finally, we can re-write the prediction of our model to be a cosine-similarity map between $f_I$ and $f_C$:

$$M_{IC}(x, y, i) = \frac{f_I(x, y) \cdot f_C(i)}{\|f_I(x, y)\| \|f_C(i)\|}, \tag{5}$$

thereby predicting masks for $f_C(i)$ being the $i$-th class feature from $f_C$, which we have obtained from clustering mask pooled features $f_M$. Accordingly, ground truth masks $M$ are also clustered according to $Q$ by converting it into hard assignment with the argmax operator [47, 22]. This can be written as $\bar{M} \in \mathbb{R}^{k \times H \times W}$ where $\bar{M}_i$ is the union of masks assigned into the cluster represented by $i$-th learned class $f_C(i)$.

**Momentum encoder for integrating mask features.** Since we jointly optimize the CLIP image encoder $\mathcal{E}_V(\cdot)$ as well as the learnable class feature $f_C$, we may experience instability during our training process, or forgetting of the pre-trained knowledge [48]. To stabilize the training, we keep a momentum encoder [39, 38] for obtaining $f_M$ as seen in Fig. 3. Therefore, we update $f_M$ as $f_M = \operatorname{MaskPool}(\bar{\mathcal{E}}_V(I), M)$, where $\bar{\mathcal{E}}_V$ is the momentum encoder of the CLIP image encoder, updated with momentum $\gamma$. This can be denoted as $\theta'_{\bar{V}} \leftarrow \gamma\theta'_{\bar{V}} + (1 - \gamma)\theta_V$, where $\theta_{\bar{V}'}, \theta_V$ are model parameters of $\bar{\mathcal{E}}_V$ and $\mathcal{E}_V$, respectively.

# 4 Experiments

## 4.1 Implementation details

For training, we employ per-pixel binary cross-entropy loss as $\mathcal{L}_{\text{mask}}$ to jointly train all of the components [6]. For all our experiments, we use a single text prompt ``A photo of {} in the scene'' for $P^*$, including for our learnable class prompts while training and for inference, we apply prompt ensemble strategy [30] with 7 additional prompts originally curated from CLIP [15]. We train our model on SA-1B [13] dataset, where we randomly sample 5% of the images. We train for 10000 iterations with a batch size of 48 for all experiments. For experiments using masks from DINO, we obtain masks with $k$-means clustering where we set $k = 16$. For experiments using masks from SAM, we use the unlabeled mask annotation in the SA-1B dataset. Without specification, we report results on ConvNeXt-B [49] backbone with mask annotation from SAM, which takes approximately 6 hours to train with 4 NVIDIA A6000 GPUs. We provide more details in the supplementary materials.

## 4.2 Experimental setting

Following Cha et al. [7], we evaluate our model on zero-shot transfer to semantic segmentation on the validation sets of COCO-Stuff [50], ADE-20K [51], PASCAL-Context [52], PASCAL VOC [53], and CityScapes [54]. For CLIP [15], we apply MaskCLIP [26] for ViT backbone for extracting image features, and remove the global pooling layer for OpenCLIP [55] with ConvNeXt [49] backbone. We note that we do not apply any post-processing to the predictions and for the compared methods. For the evaluation metric, we employ the mean Intersection over Union (mIoU).

## 4.3 Results

**Open-vocabulary semantic segmentation.** We provide results for quantitative comparisons in Tab. 1. We first compare with CLIP, and demonstrate remarkable gains in all benchmarks, bringing in an average of +16.2 mIoU improvement. Since we do not have comparable baselines without leveraging semantic labels, we further provide a comparison with image-level supervised methods [7, 9, 10]. Surprisingly, PixelCLIP surpasses TCL [7] and SegCLIP [9] in all benchmarks while using only a fraction of the images *without* semantic labels. Furthermore, we show competitive performance compared to SAM-CLIP, which uses not only 40 million image-level semantic labels, but also leverages the SA-1B dataset on a similar scale to our framework.

Table 1: **Quantitative comparison on open-vocabulary semantic segmentation.** We compare in open-vocabulary semantic segmentation with vision-language models, as well as image-level supervised methods. *: Images were seen during training. †: Masks from SA-1B [13] were used.

| Method | Training Dataset | Backbone | Additional Labels | VFM | COCO-St. | ADE-150 | Context | CityScapes | VOC |
|---|---|---|---|---|---|---|---|---|---|
| GroupViT [11] | CC12M [56], YFCC15M [57] | ViT-S/16 | - | - | 15.3 | 9.2 | 23.4 | 11.1 | 79.7 |
| CLIPpy [25] | HQITP-134M [25] | ViT-B/16 | - | - | - | 13.5 | - | - | 52.2 |
| OVSegmentor [58] | CC4M [58] | ViT-B/16 | - | DINO | - | - | 20.4 | - | 53.8 |
| CLIP [15] | WIT-400M [15] | ViT-B/16 | - | - | 16.5 | 13.2 | 25.6 | 14.9 | 73.9 |
| OpenCLIP [55] | LAION-2B [59] | ConvNeXt-B | - | - | 12.8 | 13.1 | 16.5 | 16.2 | 34.8 |
| *Training **with** additional image-level semantic labels* | | | | | | | | | |
| SegCLIP [9] | COCO [60], CC12M [56] | ViT-B/16 | Captions | CLIP | 26.5* | - | 24.7 | - | 52.6 |
| TCL [7] | CC3M, CC12M [56] | ViT-B/16 | Captions | CLIP | 19.6 | 14.9 | 30.3 | 23.1 | 77.5 |
| SAM-CLIP [10] | Merged-41M [10] | ViT-B/16 | Captions | CLIP, SAM | - | 17.1 | 29.2 | - | 60.6 |
| *Training **without** additional semantic labels* | | | | | | | | | |
| ZeroSeg [28] | ImageNet-1K [61] | ViT-B/16 | - | CLIP | 20.2 | - | 20.4 | - | 40.8 |
| PixelCLIP (Ours) | 5% SA-1B [13] (0.5M) | ViT-B/16 | - | CLIP, DINO | 22.2 | 17.4 | 34.3 | 22.9 | 83.8 |
| | | ViT-B/16 | - | CLIP, SAM† | 23.6 | 18.7 | 37.9 | 27.2 | 85.9 |
| | | ConvNeXt-B | - | CLIP, DINO | 20.2 | 19.4 | 32.7 | 30.0 | 62.9 |
| | | ConvNeXt-B | - | CLIP, SAM† | 21.4 | 20.3 | 35.4 | 34.8 | 67.2 |

Table 2: **Quantitative comparison on zero-shot mask classification.** We compare the results for mask classification using ground truth masks and generated masks from ZegFormer [1] and FC-CLIP [24]. To evaluate zero-shot mask classification from CLIP, we report the results from the zero-shot branch for both methods.

| VLM | Method | Backbone | COCO-St. | ADE-150 | Context | CityScapes | VOC |
|---|---|---|---|---|---|---|---|
| OpenCLIP [55] | Zegformer [1] | ConvNeXt-B | 15.3 | 19.1 | 24.7 | 26.5 | 51.8 |
| PixelCLIP (Ours) | Zegformer [1] | ConvNeXt-B | 23.9 (+8.6) | 21.5 (+2.4) | 38.5 (+13.8) | 34.2 (+7.7) | 71.5 (+19.7) |
| OpenCLIP [55] | FC-CLIP [24] | ConvNeXt-L | 37.3 | 27.4 | 42.8 | 35.8 | 91.4 |
| PixelCLIP (Ours) | FC-CLIP [24] | ConvNeXt-L | 46.8 (+9.5) | 30.1 (+2.7) | 52.2 (+9.4) | 48.1 (+12.3) | 90.7 (-0.7) |
| OpenCLIP [55] | Ground Truth | ConvNeXt-B | 23.8 | 30.2 | 31.4 | 32.8 | 68.3 |
| PixelCLIP (Ours) | Ground Truth | ConvNeXt-B | 34.2 (+10.4) | 34.6 (+4.4) | 51.2 (+18.4) | 41.4 (+8.6) | 85.4 (+17.1) |

**Zero-shot mask classification.** We provide results for evaluating mask classification in Tab. 2. We consider ZegFormer [1] and FC-CLIP [24] as baselines since they first predict masks, then employ CLIP as a zero-shot mask classifier within their framework, and also provide results with ground-truth masks to simulate having oracle mask predictions. For all methods, we apply masked pooling to CLIP image feature map to classify masks. For ZegFormer [1] and FC-CLIP [24], reported results are only from the zero-shot prediction branch to solely ablate our gains. We highlight that PixelCLIP can be readily applied to existing frameworks that leverage CLIP as a zero-shot mask classifier, and bring instantaneous improvements by simply replacing the model and weights of CLIP.

**Qualitative results.** We provide qualitative results for open-vocabulary semantic segmentation in Fig. 4 compared with results from CLIP, highlighting the dense open-vocabulary recognition capabilities of our framework. We further provide qualitative results in the supplementary materials.

### 4.4 Ablation studies

In Tab. 3, we show ablation studies on open-vocabulary semantic segmentation to validate our design choices. We report results without prompt ensembling for ablations, and also report results from OpenCLIP [55] as a baseline.

**Component analysis.** In Tab. 3 (a), we provide results for ablating our key components. Notably, we observe that without global semantic clustering of masks, the framework collapses and loses the pre-trained knowledge of CLIP. This validates the challenge presented by leveraging unlabeled masks and demonstrates the crucial role of our proposed clustering approach. Moreover, we observe constant improvements over all datasets with our learnable class prompt, proving our approach of leveraging the text encoder of CLIP to define the clusters in the form of prompt learning. We also observe constant gains with the momentum encoder for extracting mask pooled features $f_M$.

**Number of clusters.** In Tab. 3 (b), we compare the results of the variants of the proposed method by varying the number of clusters $k$. We find that scaling $k$ does not necessarily guarantee performance boosting, but it generally improves until $k$ is set to 64 and tends to degrade as $k$ grows. Considering that with an extremely large number for $k$, we can assign each of the masks to individual clusters(*e.g.* 1 billion for SA-1B.) This scenario would virtually be identical to not having semantic clustering as

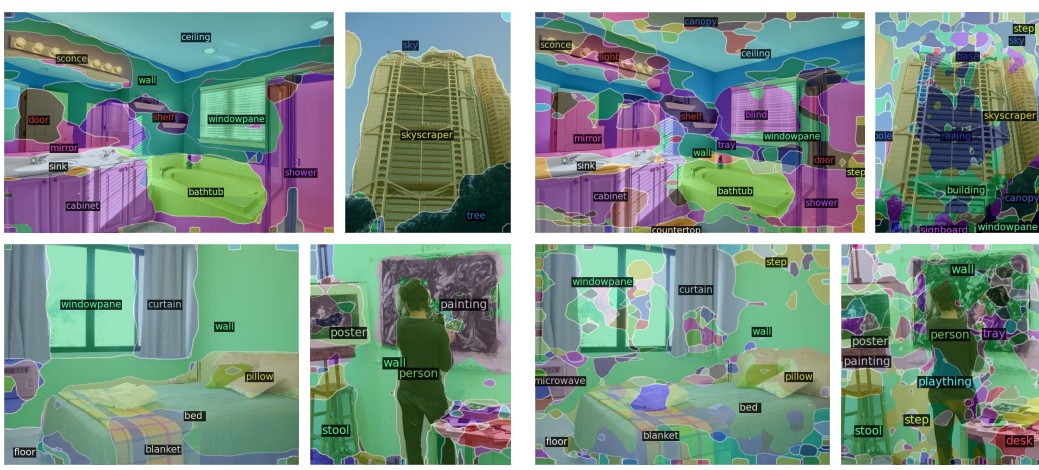

| (a) Ours | (b) CLIP |

**Figure 4: Comparison between PixelCLIP and CLIP.** We provide qualitative comparison on ADE-20K [51] dataset with PixelCLIP and CLIP. We demonstrate the dense visual recognition capabilities achieved from fine-tuning CLIP, whereas CLIP shows results with significant noise.

**Table 3: Ablation studies.** We show results on open-vocabulary semantic segmentation for validating our design choices. We also report results from OpenCLIP [55] as baseline in the results.

| Component | Evaluation Dataset | | | | |
| --- | --- | --- | --- | --- | --- |
| | COCO | ADE-150 | Context | CityScapes | VOC |
| Baseline | 12.8 | 13.1 | 16.5 | 16.2 | 34.8 |
| Ours | **21.1** | **20.2** | **34.2** | **33.2** | **66.0** |
| w/o Semantic Clustering | 0.8 | 2.1 | 4.2 | 4.4 | 6.0 |
| w/o CLIP Text Encoder | 17.9 | 18.5 | 29.9 | 28.9 | 53.5 |
| w/o Class Prompt | 18.2 | 18.8 | 30.1 | 28.1 | 54.4 |
| w/o Momentum | 19.4 | 18.5 | 28.8 | 27.2 | 58.2 |

(a) **Component analysis.** We validate the core components of our framework by ablating each components. Notably, global clustering of masks shows its importance for facilitating the framework.

| $k$ | Evaluation Dataset | | | | |
| --- | --- | --- | --- | --- | --- |
| | COCO | ADE-150 | Context | CityScapes | VOC |
| Baseline | 12.8 | 13.1 | 16.5 | 16.2 | 34.8 |
| 32 | 19.8 | 19.4 | 33.0 | 31.3 | 60.5 |
| **64** | 21.1 | 20.2 | **34.2** | **33.2** | **66.0** |
| 128 | 21.0 | 20.3 | 33.5 | 30.1 | 64.1 |
| 256 | **21.3** | **20.4** | 33.6 | 30.0 | 64.1 |
| 512 | 21.2 | 20.2 | 32.7 | 29.8 | 62.7 |

(b) **Number of clusters.** For varying $k$, we find that scaling $k$ larger than 64 does not show much improvements, while $k = 32$ also show competitive results.

| $l$ | Evaluation Dataset | | | | |
| --- | --- | --- | --- | --- | --- |
| | COCO | ADE-150 | Context | CityScapes | VOC |
| Baseline | 12.8 | 13.1 | 16.5 | 16.2 | 34.8 |
| 1 | 20.2 | 19.7 | 32.7 | 30.8 | 64.5 |
| **4** | 21.1 | **20.2** | **34.2** | 33.2 | **66.0** |
| 10 | 20.4 | 19.6 | 33.2 | 30.2 | 63.3 |
| 20 | 19.9 | 19.7 | 32.6 | **33.8** | 62.8 |

(c) **Length of learnable prompt token.** For varying $l$, we find that $l = 4$ shows best overall performance.

| Text. | Evaluation Dataset | | | | |
| --- | --- | --- | --- | --- | --- |
| | COCO | ADE-150 | Context | CityScapes | VOC |
| Baseline | 12.8 | 13.1 | 16.5 | 16.2 | 34.8 |
| Ours | **21.1** | **20.2** | **34.2** | **33.2** | **66.0** |
| COCO | 19.5 | 17.7 | 30.0 | 24.8 | 63.3 |

(d) **Effects of utilizing learnable classes.** We compare our method of learnable class prompts to having fixed set of classes from COCO-Stuff [50].

seen in Table. 3 (a), and progressively growing $k$ would slowly converge to this scenario. We further provide analysis in 4.5, studying the different aspects from varying $k$.

**Length to represent learnable class prompts.** Tab. 3 (c) compares the effects of varying the length of the learnable class prompt, $l$. We find that $l = 1$ shows lower scores in comparison to other lengths. We can interpret this as only describing a class with a single word, whereas having multiple words would better describe the depicted class. However, for $l = 4$ and larger, we find that increasing $l$ does not result in a gain of performance, hence, we adopt $l = 4$ as default.

**Effects of learnable prompt token.** Finally, we compare PixelCLIP to having a pre-defined set of classes instead of using learnable prompt tokens. Specifically, we use 171 classes from COCO-Stuff [50], and do not apply online clustering for assignment when utilizing classes from COCO-Stuff, as it already yields text features with semantic meanings. We find apparent improvements over all the datasets as shown in Tab. 3 (d). We speculate that since the classes defined in COCO-Stuff are heuristically chosen, it is hard to ideally encompass various semantics and concepts that may appear in images, hence restricting the perception of the model to the finite set of classes.

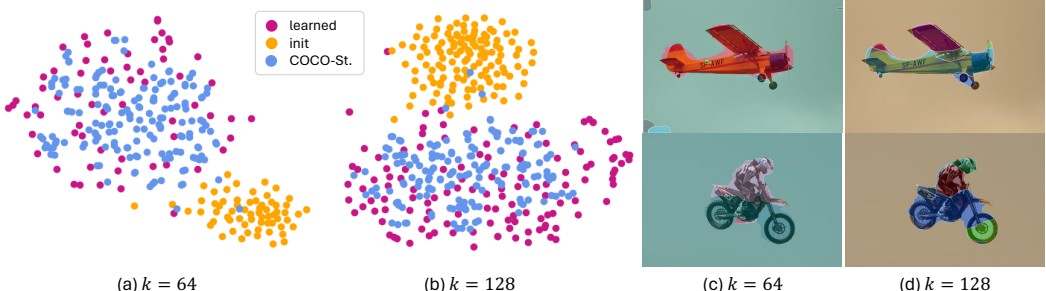

| (a) $k = 64$ | (b) $k = 128$ | (c) $k = 64$ | (d) $k = 128$ |

Figure 5: **Visualization of learned class prompts.** We visualize the text features from our learned class prompts, as well as text features from classnames of COCO-Stuff with $t$-SNE visualization in (a-b). We also visualize images inferenced with the learned class prompts in (c-d).

## 4.5 Analysis

**Learnable class prompt.** We further analyze the learned class prompt in Fig. 5 (a-b) with $t$-SNE visualization on the text features encoded from the learned class prompts, as well as text features obtained from class names of COCO-Stuff. Since we initialize the class prompt tokens as random tokens, we observe that they are in a skewed distribution in the initial state. However, the learned prompts show that they are well-dispersed among the text features from COCO-Stuff, indicating that the class prompts have well-learned diverse semantic concepts within the text features. We observe well-distributed features both for $k = 64$ and $k = 128$.

Since the learned prompts should act as implicit class names, we visualize the results from inference with learned class prompts in Fig. 5 (c-d). Although both $k = 64$ and $k = 128$ show similar performance when evaluated, we observe that the prompts have learned more fine-grained semantics for $k = 128$. We generally observe human parts to be well distinguished; this could come from the SA-1B dataset, as there are numerous images with fine-grained masks representing human parts as annotations.

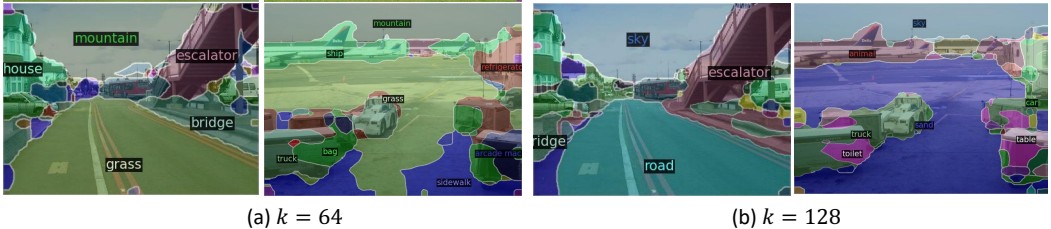

| (a) $k = 64$ | (b) $k = 128$ |

Figure 6: **Visualization of interpreting learned text prompt.** We provide visualization on results for predicting with learned class prompts, then mapping the results to classes in the dataset with the highest similarity to the prompt.

**Interpreting learned classes.** Considering the learned class prompts represent semantic concepts, we further study the learned embeddings by mapping each class embeddings to class names in COCO-Stuff with the highest cosine-similarity score. Fig. 6 shows results when we first inference the image features with learned class prompts, then map the results with the closest COCO classes. We can observe that with $k = 128$, as the prompt learns more diverse semantics, we observe more accurately mapped classes. However, we still see predictions with large disparity to the actual ground truth. We leave a more in-depth analysis of the learned classes for future investigation.

## 5 Conclusion

In this paper, we introduced PixelCLIP, a framework for leveraging unlabeled images and masks for fine-tuning the pre-trained vision-language models for open-vocabulary semantic segmentation. To address the unique challenges posed by incorporating unlabeled masks generated by vision foundation models into our framework, we propose global semantic clustering of the masks, with learnable class prompts to represent each cluster. We demonstrated PixelCLIP to show remarkable improvements to CLIP and its applicability to existing methods, providing instantaneous improvements, as well as surpassing methods that leverage image-level semantic labels such as image captions.

**Acknowledgement.** This research was supported by Institute of Information & communications Technology Planning & Evaluation (IITP) grant funded by the Korea government (MSIT) (RS-2019-II190075, RS-2024-00509279, RS-2020-II201819, RS-2024-00398115, Research on the reliability and coherence of outcomes produced by Generative AI) and the Culture, Sports, and Tourism R&D Program through the Korea Creative Content Agency grant funded by the Ministry of Culture, Sports and Tourism (RS-2024-00348469, RS-2023-00266509), and National Research Foundation of Korea (RS-2024-00346597).

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

# Appendix

## A  Further Implementation Details

We set $\gamma = 0.999$, input resolution as $H = W = 640$, which results in $h = w = 20$, and set $h' = w' = 80$ for ConvNeXt [49] backbones. For ViT [62] backbones, we set $H = W = 320$, which also results in $h = w = 20$. For global clustering, we set $\varepsilon = 1$ for ConvNeXt backbones and $\varepsilon = 0.01$ for ViT backbones. We implement our work using PyTorch [63] and Detectron2 [64]. AdamW [65] optimizer is used with a learning rate of $2 \cdot 10^{-4}$ for the decoder, $2 \cdot 10^{-5}$ for the prompt tokens and $2 \cdot 10^{-6}$ for CLIP, with weight decay set to $10^{-4}$. Prompt tokens are initialized as random word tokens with $l = 4$, and $k = 64$ as default. We use GPU implementation [66] of $k$-means clustering for our experiments with DINO masks. For $\mathcal{E}_V$, we apply CutOut [67] and color augmentations [68] during training. For the prompt ensemble strategy during inference, we use the prompts curated originally from CLIP [15] in their repository, which results in total of 8 text prompt as follows:

``itap of a {}.'',

``a bad photo of the {}.'',

``a origami {}.'',

``a photo of the large {}.'',

``a {} in a video game.'',

``art of the {}.'',

``a photo of the small {}.'',

``a photo of a {} in the scene''.

## B  Additional Experiments

Table 4: **Quantitative results with various backbones.** We show results on open-vocabulary semantic segmentation of various CLIP backbones with the addition of ViT-B from SigLIP [69] and ConvNeXt-L.

| Method | Backbone | Evaluation Dataset | | | | |
| --- | --- | --- | --- | --- | --- | --- |
| | | COCO-St. | ADE-150 | Context | CityScapes | VOC |
| SigLIP [69] | ViT-B/16 [62] | 12.4 | 11.8 | 18.3 | 19.2 | 46.8 |
| PixelCLIP (Ours) | ViT-B/16 [62] | 20.0 (+7.6) | 19.2 (+7.4) | 33.1 (+14.8) | 31.6 (+12.4) | 72.3 (+25.5) |
| CLIP [26] | ViT-B/16 [62] | 16.5 | 13.2 | 25.6 | 14.9 | 73.9 |
| PixelCLIP (Ours) | ViT-B/16 [62] | 21.4 (+4.9) | 16.7 (+3.5) | 34.9 (+9.3) | 23.8 (+8.9) | 83.1 (+9.2) |
| OpenCLIP [55] | ConvNeXt-B [49] | 12.8 | 13.1 | 16.5 | 16.2 | 34.8 |
| PixelCLIP (Ours) | ConvNeXt-B [49] | 21.1 (+8.3) | 20.2 (+7.1) | 34.2 (+17.7) | 33.2 (+17.0) | 66.0 (+31.2) |
| OpenCLIP [55] | ConvNeXt-L [49] | 16.9 | 15.2 | 22.9 | 17.1 | 57.2 |
| PixelCLIP (Ours) | ConvNeXt-L [49] | 24.8 (+7.9) | 22.6 (+7.4) | 39.4 (+16.5) | 34.3 (+17.2) | 78.9 (+21.7) |

### B.1  Results on Different Backbones

In Tab. 4, we show results for PixelCLIP when applied to different backbones. We note that since the ViT backbone has a larger output feature resolution scale compared to ConvNeXt models, we set the input image resolution to match the output feature resolution, and report results without prompt ensembling. In general, we observe noticeable gains across all backbones, with CLIP ViT-B/16 outperforming ConvNeXt-B on several datasets. Through testing with various pre-trained CLIP models, we demonstrate that our method can effectively fine-tune CLIP for dense prediction regardless of the backbone architecture.

Table 5: **Additional experiments on prompt ensembling.** We show results on open-vocabulary semantic segmentation with prompt ensembling being used during only training, only inference, or both. The default setting of prompt ensembling only being used during inference is highlighted in gray.

| Prompt Ensembling | | Evaluation Dataset | | | | |
|---|---|---|---|---|---|---|
| Training | Inference | COCO-St. | ADE-150 | Context | CityScapes | VOC |
| | | 21.4 | 16.7 | 34.9 | 23.8 | 83.1 |
| | ✓ | 23.6 (+2.2) | 18.7 (+2.0) | **37.9 (+3.0)** | 27.2 (+3.4) | **85.9 (+2.8)** |
| ✓ | | 21.6 (+0.2) | 17.1 (+0.4) | 35.1 (+0.2) | 24.9 (+1.1) | 82.9 (-0.2) |
| ✓ | ✓ | **23.7 (+2.3)** | **19.2 (+2.5)** | **37.9 (+3.0)** | **28.1 (+4.3)** | 85.5 (+2.4) |

## B.2   Analysis on Prompt Ensembling

In Tab. 5, we show results with prompt ensembling being applied during only training, only inference, and both. We report results with ViT-B/16 using SA-1B masks as supervision. Although prompt ensembling does bring slight gains when enabled during training, the computation for optimizing learnable class prompts scales along with the number of prompts used, increasing the training time and the memory consumption. On the other hand, applying prompt ensembling during inference only adds negligible cost as they can be computed once and be cached, but shows much significant gains compared to when applied training. Therefore, we adopt prompt ensemlbing only during inference, but noticing that the performance can be maximized with better prompts during training. In this regard we can better results with better prompt design or a learnable prefix to accompany the learnable class prompts, which we leave for future investigation.

## B.3   Additional Ablation Studies

Table 6: **Additional ablation studies**. We show results on open-vocabulary semantic segmentation with a larger number of clusters and different training datasets.

| $\gamma$ | Evaluation Dataset | | | | |
|---|---|---|---|---|---|
| | COCO | ADE-150 | Context | CityScapes | VOC |
| Baseline [55] | 12.8 | 13.1 | 16.5 | 16.2 | 34.8 |
| 0.99 | 19.9 | 19.5 | 32.5 | 29.5 | 62.6 |
| **0.999** | **21.1** | **20.2** | **34.2** | **33.2** | **66.0** |
| 0.9999 | 20.4 | 19.7 | 32.0 | 29.9 | 63.0 |

(a) **Varying momentum $\gamma$.** We show additional results for varying $\gamma$ for the momentum update.

| Dataset | Evaluation Dataset | | | | |
|---|---|---|---|---|---|
| | COCO | ADE-150 | Context | CityScapes | VOC |
| Baseline [55] | 12.8 | 13.1 | 16.5 | 16.2 | 34.8 |
| COCO-St. [50] | **24.1** | **21.9** | **36.8** | 30.2 | **71.0** |
| SA-1B [13] | 21.1 | 20.2 | 34.2 | **33.2** | 66.0 |

(b) **Different training dataset.** We show results for leveraging ground-truth masks from COCO-Stuff while removing its class labels.

### B.3.1   Ablation on the momentum update rate $\gamma$

In Tab. 6 (a), we show results for varying $\gamma$ for the momentum update. While having the momentum encoder generally shows improvements, we find $\gamma = 0.999$ to show the best results for updating the momentum encoder.

### B.3.2   Ablation on training dataset

In Tab. 6 (b), we show results for training with mask annotation from COCO-Stuff [50]. For COCO-Stuff, we remove the ground truth class labels and utilize them as unlabeled masks, and other hyperparameters are set identically with $k = 64$. Although the masks from COCO-Stuff show better results across all datasets, we highlight that the SA-1B [13] dataset mostly consists of automatically generated masks from SAM, whereas COCO-Stuff has human annotated masks from expert annotators.

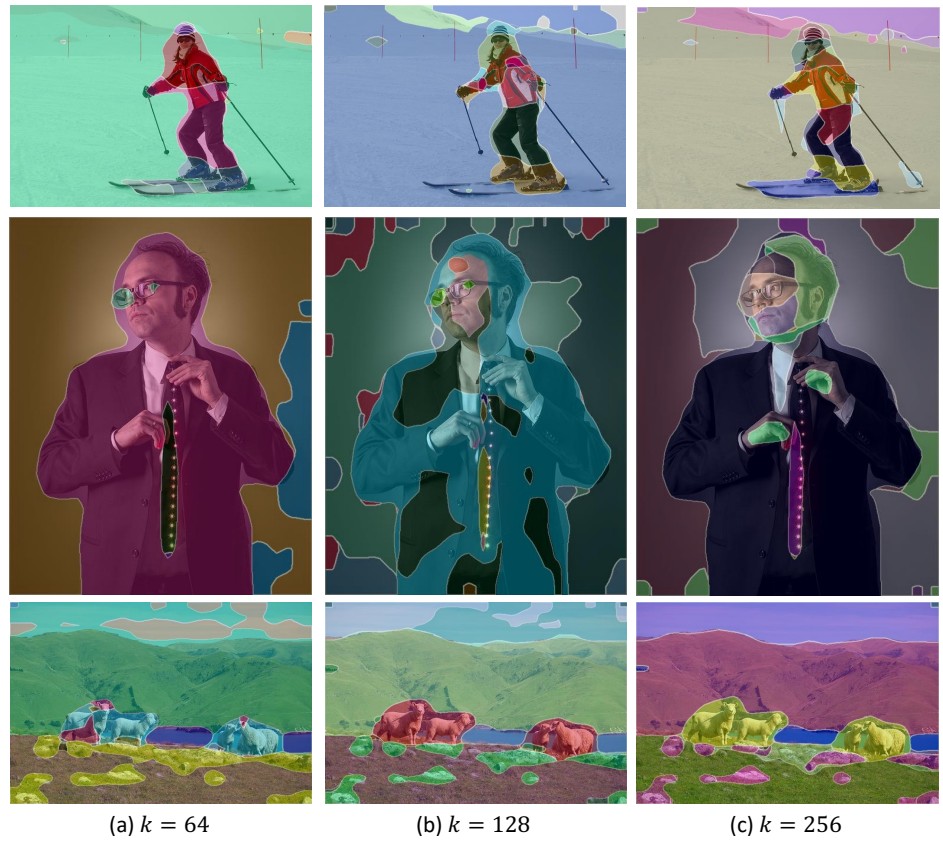

(a) $k = 64$        (b) $k = 128$        (c) $k = 256$

Figure 7: **Visualization on COCO-Stuff with learned class prompts.** We provide results with learned classes with different $k$ up to 256.

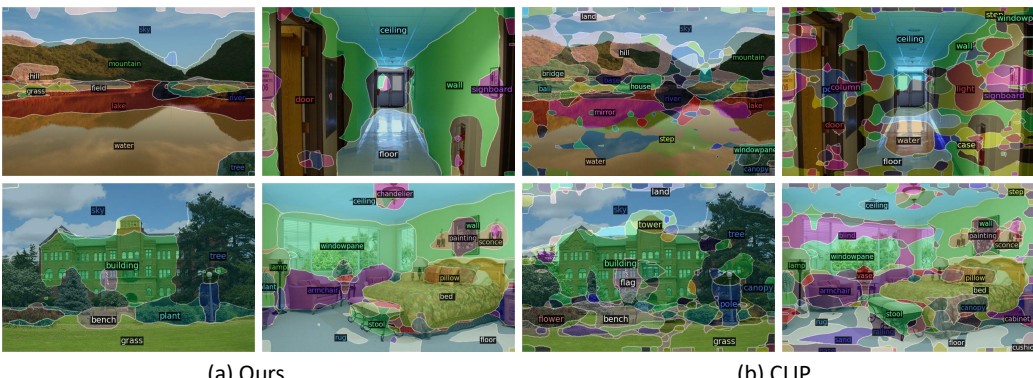

(a) Ours                                     (b) CLIP

Figure 8: **Visualization on ADE-20k** We provide qualitative comparison on ADE-20K [51].

## C   Additional Qualitative Results

We provide qualitative results of visualization on ADE-20K [51], PASCAL-Context [52] in Fig. 8 and Fig. 9.

## D   Additional Visualization

In Fig. 7, we show visualization on COCO-Stuff by classifying the image features with our learned class prompts for varying $k$. From the first and the second row, we can observe that with larger numbers of $k$, different parts of human are segmented into fine-grained regions whereas $k = 64$ has

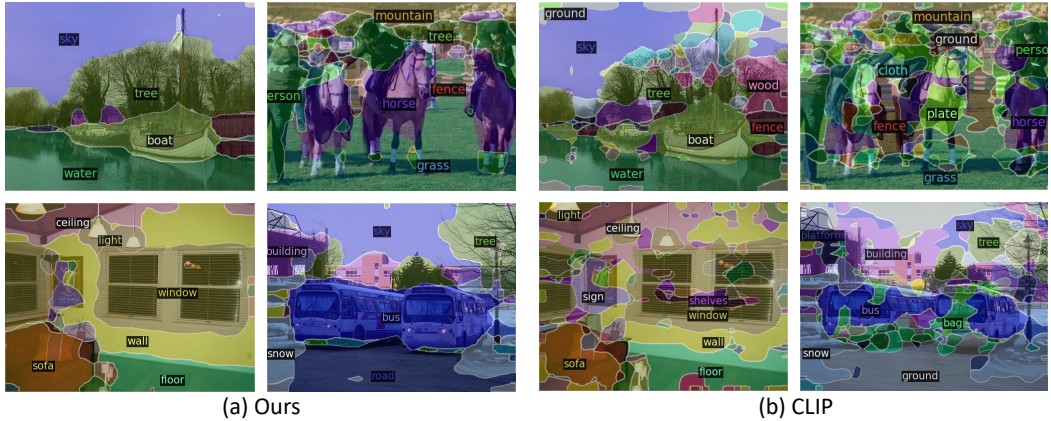

|(a) Ours|(b) CLIP|

Figure 9: **Visualization on PASCAL-Context.** We provide qualitative comparison on PASCAL-Context [52].

more coarse regions. Especially for $k = 256$, in the second row, we observe the glasses, hair, and hands all classified into different classes with our learned prompt.

On the other hand, we also observe cases where a small number of $k$ struggles to differentiate visual concepts in the last row, where the animals are partially grouped with $k = 64$ and show better groups for $k = 128$ and $k = 256$. This could indicate that with only a small number of clusters, several fine-grained visual concepts that may not be seen often in the dataset to be grouped as a whole, whereas independent clusters could be assigned with a larger number of $k$, allowing fine-grained recognition of semantics.

## E  Limitations

Although we aim to fine-tune the image encoder of CLIP for adapting to dense predictions, we initialize the mask encoder within our framework with pre-trained weights of CLIP, which yields poor results for classifying masks when applying mask pooling to its features. Consequently, the noisy mask features in the earlier stage of training may result in sub-optimal performance. While there could be alternative methods to extract per-mask CLIP image features, we consider mask pooling to be sufficient to show meaningful improvements to CLIP and consider such exploration for future directions.

## F  Broader Impact

Our framework facilitates open-vocabulary semantic segmentation through leveraging vision-language models, hence the recognition capabilities of our method rely on the pre-trained knowledge of the vision-language models. Considering that large-scale pre-trained vision-language models [15, 16, 55] leverage web-crawled data within its training, the models may exhibit wrongful behaviors from bias or corrupted data from the internet which calls for future research to address.

