# OpenReview forum: "Towards Open-Vocabulary Semantic Segmentation Without Semantic Labels"
_NeurIPS.cc/2024/Conference — NeurIPS 2024 poster_

### Official Review · Reviewer_83hz · 2024-07-06

**Soundness:** 3
**Presentation:** 3
**Contribution:** 2
**Rating:** 5
**Confidence:** 4

**Summary:**

This paper present MCLIP, which aims to finetune CLIP using image and mask data without semantic labels. The goal is to make its open-vocabulary recognition ability to be adapted to position-sensitive semantic segmentation tasks. MCLIP chooses to use SAM or feature clusters of DINO to obtain masks, which are class-agnostic but sometimes too small to have consistent semantics for finetuning. In addition, in order to achieve a trade-off between training stability and avoiding catastrophic forgetting, clustering uses the EMA version of finetuned image encoder.

**Strengths:**

- The idea of this paper is interesting. Solving open-vocabulary semantic segmentation without semantic labels is an interesting topic.

- The experimental results show that the preformance of the proposed approach is good.

- Ablation studies and analysis are provided to help better understand of the role of each component and hyper-parameter.

**Weaknesses:**

- What role does the text encoder play in the clustering process?

The key to the success of clustering lies in the existence of unlabeled data in a certain grouping relationship in the space. The mask representations f_M can be considered to meet such conditions. If k learnable object prompts are initialized randomly, then k text representations f_C are randomly scattered in the text space initially, unable to provide the information needed for clustering. If k learnable object prompts are initialized by predefined categories, then this should violate the experimental setup without semantic labels. To illustrate this problem, the authors need to provide an explanation and add a comparative experiment: What results can be obtained just by clustering with f_M?

- Why use clustering technology to solve the problem of too fine initial mask granularity?

Firstly, previous work (GroupViT, CVPR'22 and SegCLIP, ICML'23) has used a grouping block designed based on the idea of clustering in the process of fine-tuning CLIP. In contrast, the clustering used in MCLIP provides better masks initially than the image patches used in previous work and the clustering process of MCLIP is parameter-free. It may be necessary to conduct ablation experiments to prove which differences bring positive results. But overall, technically, the use of clustering may be not novel.

- In order to obtain class-agnostic masks, there is a class of experimental settings called open set semantic
segmentation (O3S, arXiv preprint arXiv:2307.02003, 2023) . Can the ”SAM initialization + clustering” scheme proposed by MCLIP obtain competitive performance in the O3S experimental setting?

- Table 1 shows that using MCLIP’s fine-tuned CLIP instead of the Frozen CLIP in the previous method improves the performance of the previous method’s zero-shot mask classification. Taking FC-CLIP as an example, MCLIP improves its zero-shot mask classification performance on ADE20K by 2.7. But compared to the final performance of FC-CLIP with two branches integrated, it is still 4.0 lower. So, in the case of keeping the complete inference strategy of the previous method, how much improvement can be achieved by using MCLIP’s fine-tuned CLIP instead of the Frozen CLIP?

- In the ablation experiment in Table 3a, does ’w/o Semantic Clustering’ mean fine-tuning with SAM-initialized masks directly without clustering? If so, why is the performance so poor? Without using clustering, it is equivalent to the number of cluster k is greater than the number of masks initialized by SAM on all images. However, the conclusion obtained from table 3b is that with the increase of k, the
performance tends to saturation, rather than decreasing to the level of ”w/o Semantic Clustering”.

**Questions:**

I think the authors need to further explain the novelty of this paper and give more analysis on the experimental results as listed in the weaknesses part.

**Limitations:**

Limitations are provided. It seems that there is no potential negative societal impact.

---

> ### Author Rebuttal · Authors · 2024-08-07
>
> We thank reviewer 83hz for the remarks on the paper and the considerate review. We address the comments and questions from the reviewer below:
>
> > **What role does the text encoder play in the clustering process?**
> >
>
> |  | COCO | ADE-150 | PC-59 | Cityscapes | VOC |
> | --- | --- | --- | --- | --- | --- |
> | without Text enc. | 17.9 | 18.5 | 29.9 | 28.9 | 53.5 |
> | with Text enc. | 21.2 | 20.2 | 34.2 | 33.2 | 66.0 |
>
> *Table G. Ablation on the usage of text encoder for clustering during training.*
>
> We use the text encoder to match the inference strategy where classes are passed through the text encoder, as well as for leveraging its pre-trained knowledge to aid the semantic clustering process. The reviewer is correct in that the clustering process is possible without the text encoder by initializing $f_C$ as random vectors. We show such ablations in Table G where we verify that although the clustering is possible without the text encoder, it largely benefits with the text encoder. We thank the reviewer for pointing out an important factor for ablation, and we will add the results and discussions to the paper.
>
> > **Why use clustering technology to solve the problem of too fine initial mask granularity?**
> >
>
> We would like to highlight that our clustering is done at a global level: we cluster masks across all images in the batch. GroupViT and SegCLIP in contrast cluster at a local level: they cluster regions within a single image using the caption for that image. Therefore, it is not obvious how we can use the GroupVit or SegCLIP grouping blocks in our framework. We will update the Related Work in our revision with this discussion. We emphasize that although the use of clustering is not novel, to the best of our knowledge,  our use of clustering across images for unsupervised representation learning is.
>
> > **Can the ”SAM initialization + clustering” scheme proposed by MCLIP obtain competitive performance in the O3S experimental setting?**
> >
>
> |  | Fold-0 | Fold-1 | Fold-2 | Fold-3 | mean |
> | --- | --- | --- | --- | --- | --- |
> | ZS3[3] | 18.8 | 20.1 | 24.8 | 20.5 | 21.1 |
> | LSeg[b] | 22.1 | 25.1 | 24.9 | 21.5 | 23.4 |
> | Fusioner[c] | 23.6 | 28.2 | 26.2 | 24.1 | 25.5 |
> | Yang et al.[d] | 26.5 | 30.8 | 26.3 | 24.1 | 26.9 |
> | MCLIP | 25.1 | 25.5 | 22.1 | 24.1 | 24.2 |
>
> *Table H. Results on COCO-20i under Z/FS setting from Yang et al.[d] Following LSeg[b], we used ‘others’ for the background class and prompt ensembling was used during inference.*
>
> To validate the performance in O3S setting, we directly evaluate MCLIP trained on SA-1B on COCO-20i without training with any COCO images or annotations. Interestingly, we find MCLIP to show reasonable performance, surpassing LSeg despite not being trained on other folds of the COCO-20i dataset. We thank the reviewer for suggesting the evaluation, and we will add it to our revision.
>
> > **So, in the case of keeping the complete inference strategy of the previous method, how much improvement can be achieved by using MCLIP’s fine-tuned CLIP instead of the Frozen CLIP?**
> >
>
> |  | COCO-Panoptic | ADE-150 | PC-59 | Cityscapes | VOC |
> | --- | --- | --- | --- | --- | --- |
> | FC-CLIP | 63.7 | 34.1 | 58.4 | 56.2 | 95.4 |
> |  + MCLIP | 64.9 | 34.1 | 59.2 | 56.8 | 95.6 |
>
> Table I. Results from FC-CLIP with the complete inference strategy
>
> We report the results with the full pipeline from FC-CLIP, when incorporating the “out-of-vocabulary” branch with MCLIP in Table I. We notice that due to the fusion strategy, the gains from the “out-vocabulary” branch are not completely transferred to the fused scores, but observe improvements over all datasets. We will add these results to the revision.
>
> > **Why is the performance so poor without Semantic Clustering? The conclusion obtained is that with the increase of k, the performance tends to saturation rather than decreasing to the level of ”w/o Semantic Clustering”.**
> >
>
> | Number of clusters, k | COCO | ADE-150 | PC-59 | Cityscapes | VOC |
> | --- | --- | --- | --- | --- | --- |
> | 64 | 21.1 | 20.2 | 34.2 | 33.2 | 66.0 |
> | 128 | 21.0 | 20.3 | 33.5 | 30.1 | 64.1 |
> | 256 | 21.3 | 20.4 | 33.6 | 30.0 | 64.1 |
> | 512 | 21.2 | 20.2 | 32.7 | 29.8 | 62.7 |
>
> *Table J. Additional results for increasing the number of clusters k. Performance starts decreasing after k=256*
>
> We clarify that ‘w/o Semantic Clustering’ refers to supervising CLIP directly with fine-grained masks from SAM, i.e. f_C=f_M, which is to the reviewer’s understanding. This would enforce CLIP to distinguish small regions within the image, despite having near identical semantic meanings. We find this to conflict with the coarse, semantic understanding of CLIP when fine-tuning, eventually losing the pre-trained knowledge of CLIP, i.e. catastrophic forgetting, which is critical when fine-tuning foundation models[11, 43, 65].
>
> Furthermore, as the reviewer pointed out, ‘w/o Semantic Clustering’ would virtually be k=50M, (5% of SA-1B), and we agree that further increasing k should start to decline the performance, eventually reaching 50M. To verify this, we additionally provide results with k=512 in Table J which is the maximum number of k within our GPU memory constraints, and observe the performance decreasing for all datasets. We greatly thank the reviewer for providing the insight, and we will add the discussions to the revision.
>
> References:
>
> [b] Language-driven Semantic Segmentation. B. Li et al., ICLR 2022
>
> [c] Open-vocabulary Semantic Segmentation with Frozen Vision-Language Models, C. Ma et al., BMVC 2022
>
> [d] Multi-Modal Prototypes for Open-World Semantic Segmentation, Y. Yang et al., IJCV 2024

---

> > ### Comment · Reviewer_83hz · 2024-08-12
> > **Final rating**
> >
> > I appreciate the responses from the reviewers. Though most of my concerns have been solved, I still think the novelty of this paper cannot reach the level of an accept. So, I tend to keep my original rating unchanged.

---

### Official Review · Reviewer_kepm · 2024-07-08

**Soundness:** 3
**Presentation:** 3
**Contribution:** 3
**Rating:** 6
**Confidence:** 4

**Summary:**

The authors introduce a novel unsupervised formulation of open vocabulary semantic segmentation, adapting a pre-trained vision-language model (CLIP) to the task via distillation from vision-only segmentation models (i.e., SAM, DINO). To use the language encoder without constraining fine-tuning to the set of classes of the dataset, they propose to apply online clustering at the dataset level and to learn class-specific feature prototypes. The method is compared against state-of-the-art approaches trained with and without supervision, and the results validate the approach as a practical direction for training open vocabulary models for semantic segmentation.

**Strengths:**

- The work is well-presented and curated, and the motivation is clear and sound.
- The approach finetunes models pre-trained for a different purpose, and employs supervision from other pre-trained models, effectively reusing knowledge efficiently. Distillation from pre-trained models also removes the need for data supervision, potentially permitting scale training for open vocabulary semantic segmentation to billions of samples.
- The method is tested against various baseline and on different benchmark datasets.
- The model components are evaluated independently in the ablation study, helping to uncover the individual contribution to the final picture.

**Weaknesses:**

- While I understand the rationale behind performing clustering at pixel level on the entire dataset, I am not sure of the scalability of the approach. This probably explains why the authors only use 5% of the SA-1B dataset. It would be helpful to quantify the costs of performing the online clustering at training time.
- Since the method could potentially suffer scalability issues, it would be interesting to understand performance when trained unsupervised in-distribution, reporting both in- and out-of-distribution performance, e.g., MCLIP trained on VOC and tested on Context.
- The reasoning behind the selection process for the prompt (i.e., "a photo of a {} in the scene") is unclear. This is a minor issue, but the selection should probably be justified. While I would expect the model to learn to ignore the prefixes and suffixes, it would be interesting to understand how the model performs with other prompts.
- Using CLIP as a baseline for the ablation studies may be misleading due to the mismatch between its application and open vocabulary semantic segmentation

**Questions:**

- What are the costs of performing online clustering at the dataset level during training?
- What is the performance of MCLIP when trained on the benchmark datasets and tested in- and out-of-distribution? How does it compare against some baseline methods?
- What is the model performance when changing the template at inference time? What is the performance without the template? What about training without the template? Does performance improve with prompt ensembling (i.e., similar to what CLIP does with N prompts that average to get better class centroids)?

**Limitations:**

- A potential limitation is the one I reported above, i.e., scalability issues due to the global online clustering procedure.

---

> ### Author Rebuttal · Authors · 2024-08-07
>
> We thank reviewer kepm for the remarks on the paper and the considerate review. We address the comments and questions from the reviewer below:
>
> > **It would be helpful to quantify the costs of performing the online clustering at training time. What are the costs of performing online clustering at the dataset level during training?**
> >
>
> | Training dataset | COCO | ADE-150 | PC-59 | Cityscapes | VOC |
> | --- | --- | --- | --- | --- | --- |
> | 5% SA-1B | 21.4 | 16.7 | 34.9 | 23.8 | 83.1 |
> | 10% SA-1B | 21.6 | 16.9 | 36.1 | 23.9 | 83.9 |
>
> *Table C. Results on open-vocabulary semantic segmentation with additional training data.*
>
> We compute the cluster assignment through the Sinkhorn-Knopp algorithm [13], which only adds around 1ms in our training every iteration, as it can be performed on GPU. We opted for a 5% split of SA-1B due to the large scale of the full dataset, which is similar in scale with SAM-CLIP [43]. To study the scalability of our approach, we provide results when doubling the amount of training data in Table C, showing that we do observe modest improvements from more training data. We will add this experiment to the revision.
>
>
> > **What is the performance of MCLIP when trained on the benchmark datasets and tested in- and out-of-distribution? How does it compare against some baseline methods?**
> >
>
> |  | COCO-Stuff | ADE-150 | PC-59 | Cityscapes | VOC |
> | --- | --- | --- | --- | --- | --- |
> | MaskCLIP[61] | 16.5 | 13.2 | 23.4 | 11.1 | 79.7 |
> | MaskCLIP+[61] | 18.0* | - | 31.1* | - | - |
> | MCLIP (COCO+DINO) | 20.0* | 15.5 | 30.1 | 18.5 | 79.4 |
> | MCLIP (COCO+SAM) | 21.7* | 16.6 | 34.2 | 23.1 | 82.4 |
>
> *Table D. Results on open-vocabulary semantic segmentation for in- and out-of-distribution analysis. \*: indicates in-distribution results where the training splits of the datasets were seen.*
>
> We thank the reviewer for providing an interesting point for discussion. To study in- and out-of-distribution, we provide “in-distribution” results by training MCLIP with COCO-Stuff images, and “out-of-distribution” results by zero-shot evaluating on other datasets. For comparison, we provide MaskCLIP as “out-of-distribution” and MaskCLIP+ suggested by reviewer Dx6v as “in-distribution” baselines, where MaskCLIP+ is trained with images from the target dataset COCO-Stuff and PC-59 respectively. We observe that our MCLIP outperforms MaskCLIP+ in COCO-Stuff, but the in-distribution performance of MaskCLIP+ outperforms MCLIP in PC-59 with DINO masks. However, when incorporating stronger SAM masks, MCLIP can improve largely, outperforming MaskCLIP+ despite being out-of-distribution for PC-59. We will add this experiment and discussion to the revision.
>
> > **What is the model performance when changing the template at inference time? What is the performance without the template? What about training without the template?**
> >
>
> | Training(↓)/Inference(→) | {} | itap of a {} | a photo of a {} | a photo of a {} in the scene |
> | --- | --- | --- | --- | --- |
> | {} | 35.4 | 38.7 | 38.0 | 36.6 |
> | itap of a {} | 35.4 | 39.2 | 38.1 | 36.3 |
> | a photo of a {} | 35.3 | 38.6 | 37.8 | 36.4 |
> | a photo of a {} in the scene | 35.1 | 38.6 | 37.8 | 36.0 |
>
> *Table E. Results on open-vocabulary semantic segmentation with different prompts. Each row show results from different prompts in training, while each column is in inference. For brevity, we report the mean over 5 benchmarks from Table D. “itap” is a common abbreviation of “I took a picture”*
>
> We thank the reviewer for suggesting ablation with different prompts. We provide results when trained with no prompt(“{}”)  and with different prompts in Table E. We initially used “a photo of a {} in the scene” following other methods [11, 14, 21] but surprisingly, the prompt “itap of a {}” shows significant improvements for both training and inference. Given that “itap of a {}” is one of the well-performing prompts originally curated from CLIP, we speculate the results to reflect the preference of prompts from CLIP. We will add the results and the discussions, and re-conduct our experiments with better prompts.
>
> > **Does performance improve with prompt ensembling (i.e., similar to what CLIP does with N prompts that average to get better class centroids)?**
> >
>
> | Training | Inference | COCO-Stuff | ADE-150 | PC-59 | Cityscapes | VOC |
> | --- | --- | --- | --- | --- | --- | --- |
> | No ensemble | No ensemble | 21.4 | 16.7 | 34.9 | 23.8 | 83.1 |
> |  | Ensemble | 23.6 (+2.2) | 18.7 (+2) | 37.9(+3.0) | 27.2(+3.4) | 85.9(+2.8) |
> | Ensemble | No ensemble | 21.6(+0.2) | 17.1(+0.4) | 35.1(+0.2) | 24.9(+1.1) | 82.9(-0.2) |
> |  | Ensemble | 23.7(+2.3) | 19.2(+2.5) | 37.9(+3.0) | 28.1(+4.3) | 85.5(+2.4) |
>
> *Table F. Results on open-vocabulary semantic segmentation with prompt ensembling strategy.*
>
> We provide results for ensembling in Table F, where we ensemble the default prompt “a photo of a {} in the scene” with 7 additional prompts originally curated in CLIP. We observe that not only can prompt ensembling largely boost the performance during inference time, it also shows additional gains when applied during training. We thank the reviewer for the valuable suggestion, and we will add the results and discussion to the paper.
>
> > **Using CLIP as a baseline for the ablation studies may be misleading due to the mismatch between its application and open vocabulary semantic segmentation**
> >
>
> We clarify that the baseline results from CLIP in the ablation studies are in fact obtained from applying MaskCLIP [61], which slightly modifies CLIP for open-vocabulary semantic segmentation. We apologize for the confusion, and we will revise the baseline as MaskCLIP in the ablations.

---

> > ### Comment · Reviewer_kepm · 2024-08-08
> >
> > I thank the authors for their reply, which answers my doubts and questions. I appreciate that they executed all the experiments I proposed. Scaling the dataset size seems to (unsurprisingly) improve model performance. Also, a better template for the input query or a template ensemble has a positive impact. For the first, I believe the work would benefit from some experiments showing how performance increases with an increase in the number of train samples. I expect the performance to improve until the compute-optimal point is reached [1].
> >
> > Overall, I am glad my suggestions improved the method's performance, however, this also gives the impression the work was not carried out very rigorously. In any case, I am satisfied with the authors' rebuttal and confirm my initial positive rating.
> >
> > [1] Hoffmann, Jordan, et al. "Training compute-optimal large language models." arXiv (2022).

---

### Official Review · Reviewer_rHvB · 2024-07-14

**Soundness:** 2
**Presentation:** 3
**Contribution:** 2
**Rating:** 5
**Confidence:** 4

**Summary:**

This paper proposes to learn an open-vocabulary semantic segmentation model with only unlabeled images and pretrained foundation models, such as SAM and DINO. The intuition is that CLIP model already knows what is in the image, so we only need to teach CLIP where the object is. It first uses pretrained DINO to generate pseudo masks and then exploits a online clustering method to group the part segments into valid object masks. It also proposes learnable class embeddings to solve the problem of lacking ground-truth text labels. Compared with baselines, the proposed method achieves decent improvements.

**Strengths:**

+ This paper proposes a solution to train an open-vocabulary model only with unlabeled images, i.e., without masks nor captions.

+ Three technical contributions to make the solution happen: (1). use DINO to generate pseudo masks (2). group part masks into objects (3). user learnable embeddings to substitute text captions.

**Weaknesses:**

- Compared with other methods  [ 50 , 30 , 43 , 51 ] leveraging image caption as supervision, this paper actually uses stronger DINO generated pseudo masks to train the segmentation model. Furthermore, it even uses the SAM masks during experiments (In Table 2). Regrading this, I think this method includes strong segmentation prior into the training, making the comparison unfair.

- Compared with methods using similar VFM, such as SAM-CLIP, the proposed method performs much worse. I understand it is not an apple-to-apple comparison due to the differences in training data. More specially, this paper doesn't use captions. However, caption is also very easy to get with pretrained caption models. If simply adding caption could boost the performance so much, why should we stick to a setting without caption?

**Questions:**

- Why does Table 1 appear before Table 2? Tabel 1 looks more like an ablation study to me.

**Limitations:**

See weakness

---

> ### Author Rebuttal · Authors · 2024-08-07
>
> We thank reviewer rHvB for the remarks on the paper and the considerate review. We address the comments and questions from the reviewer below:
> > **Compared with methods using similar VFM, such as SAM-CLIP, the proposed method performs much worse.**
> >
>
> |  | COCO-Stuff | COCO-Object | ADE-150 | PC-59 | Cityscapes | VOC |
> | --- | --- | --- | --- | --- | --- | --- |
> | GroupViT[50] | 15.3 | 27.5 | 9.2 | 23.4 | 11.1 | 79.7 |
> | SAM-CLIP[43] | - | 31.5 | 17.1 | 29.2 | - | 60.6 |
> | MCLIP (DINO) | 20.2 | 41.9 | 15.6 | 31.3 | 19.2 | 80.0 |
> | MCLIP (SA-1B) | 21.4 | 43.3 | 16.7 | 34.9 | 23.8 | 83.1 |
>
> *Table B. Additional comparison on open-vocabulary semantic segmentation including COCO-Object, evaluated on 80 instance classes of COCO-Stuff.*
>
> Thank you. We would first like to point out that SAM-CLIP actually reported results on COCO-Object and not COCO-Stuff, which we have verified with the authors of SAM-CLIP. We therefore present the corrected comparisons in Table B, which we will also include in the revision. We can see that our proposed MCLIP does indeed outperform SAM-CLIP on COCO-Object, consistent with PC-59 and VOC, and is only marginally behind by 0.4 points on ADE. We achieve this strong performance despite SAM-CLIP additionally using 40M image-text pairs, showing the benefits of our approach.
>
> > **If simply adding caption could boost the performance so much, why should we stick to a setting without caption?**
> >
>
> We would like to highlight the main contribution to be our exploration for effectively leveraging unlabeled masks, which has also been acknowledged by other reviewers to be a “neat idea” (Dx6v), “motivation is clear and sound” (kepm) and “interesting” (83hz), hence focus on masks instead. The framework can be potentially enhanced by incorporating captions along with unlabeled masks, which we anticipate for future exploration.
>
> > **Compared with other methods [ 50 , 30 , 43 , 51 ] leveraging image caption as supervision, this paper actually uses stronger DINO generated pseudo masks**
> >
>
> We emphasize that we do not need human labels within our training as DINO is trained in a self-supervised manner with only images. In contrast, the mentioned methods leverage human-annotated captions in addition to images, which makes it hard to consider that our method is leveraging stronger supervision. Furthermore, we also highlight that SAM is leveraged to demonstrate scenarios with higher quality unlabeled masks, and MCLIP still shows strong performance with DINO generated masks.
>
> > **Table 1 looks more like an ablation study to me.**
> >
>
> We thank the reviewer for the suggestion, and will adjust the tables accordingly.

---

> > ### Comment · Reviewer_rHvB · 2024-08-07
> > **I've raised my score from 4 to 5**
> >
> > Thanks for the rebuttal! I've raised my score from 4 to 5.

---

### Official Review · Reviewer_Dx6v · 2024-07-15

**Soundness:** 3
**Presentation:** 3
**Contribution:** 2
**Rating:** 6
**Confidence:** 4

**Summary:**

This paper proposes to enhance the semantic segmentation performance of the pretrained CLIP model using unlabeled images and pseudo segmentation masks generated with vision foundation models such as SAM and DINO. Specially, the pseudo masks are acquired via a online feature clustering algorithm. Experiments on standard benchmarks demonstrate the superior performance over CLIP and competitive results to existing open-vocabulary semantic segmentation methods.

**Strengths:**

1. It is a neat idea to leverage unlabeled masks  as supervision generated from foundation models (e.g. SAM, DINO) for the open-vocabulary semantic segmentation task.
2. The paper is well-written and easy to follow.
3. The ablation study demonstrates the effectiveness of some design choices such as momentum update of the image encoder, online cluster assignment, and learning class prompts. Performance on the standard benchmarks also show the competitive results in comparison with existing baseline methods.

**Weaknesses:**

1. How is the proposed method in comparison with MaskCLIP+? Figure 4 shows the visual comparisons between the proposed method and MaskCLIP, as the Sec. 4.2 mentioned “For evaluating CLIP [42, 10], we apply MaskCLIP [61] to extract CLIP image features”. It would be interesting to compare the proposed method with MaskCLIP+ which distills MaskCLIP to train more advanced segmentation model.
2. Some related works should be discussed and compared to the proposed method, for example, Exploring Open-Vocabulary Semantic Segmentation from CLIP Vision Encoder Distillation Only. ICCV 2023.

**Questions:**

See the weakness section.

**Limitations:**

Yes

---

> ### Author Rebuttal · Authors · 2024-08-07
>
> We thank reviewer Dx6v for the remarks on the paper and the considerate review. We address the comments and questions from the reviewer below:
> > **Comparison with MaskCLIP+**
> >
>
> |  | COCO-St. | ADE-150 | PC-59 | Cityscapes | VOC |
> | --- | --- | --- | --- | --- | --- |
> | MaskCLIP+[61] | 18.0 | - | 31.1 | - | - |
> | ZeroSeg[a] | 20.2 | - | 20.4 | - | 40.8 |
> | MCLIP (ours) | 21.4 | 16.7 | 34.9 | 23.8 | 83.1 |
>
> *Table A. Additional comparison on open-vocabulary semantic segmentation with other methods.*
>
> We provide comparison with MaskCLIP+ in Table A, with results from ViT-B/16 CLIP backbone. Despite MaskCLIP+ having an advanced decoder for segmentation and requiring training on the target dataset, we demonstrate that MCLIP outperforms MaskCLIP+ in both COCO-Stuff and PC-59, demonstrating the effectiveness of our approach.
>
> > **Some related works should be discussed and compared**
> >
>
> We thank the reviewer for pointing out related works with our work. We provide results from ZeroSeg[a] as mentioned by the reviewer in Table A, and we will add discussions and comparison to ZeroSeg and MaskCLIP+ to the paper.
>
> References:
>
> [a] Exploring Open-Vocabulary Semantic Segmentation from CLIP Vision Encoder Distillation Only. J. Chen et al., ICCV 2023

---

> > ### Comment · Reviewer_Dx6v · 2024-08-12
> >
> > Thanks for the detailed response! I'll raise my initial rate to weak accept.

---

### Author Rebuttal · Authors · 2024-08-07

We thank the reviewers for the remarks on the paper, as well as their considerate reviews. Especially, we appreciate the thoughtful comments on the idea and motivation for leveraging unlabeled masks (**Dx6v, kepm, 83hz**), paper being well-written and curated (**Dx6v, kepm**), the proposed solution and technical contributions (**rHvB**), solid experiments and ablations (**Dx6v, kepm, 83hz**), as well as our approach effectively reusing knowledge from other pre-trained models (**kepm**).

Furthermore, we thank the reviewers for sharing insights and providing constructive feedback, which we have responded below. We hope our response adequately addresses the concerns, and we believe that the addition of the discussions will greatly improve the paper.

---

### Decision · Program_Chairs · 2024-09-25

**Decision:**

Accept (poster)

**Comment:**

All reviewers found the article interesting, exploring the use of unlabeled masks for building an open vocabulary segmentation model. In general, reviewers found that the results and analyses support the proposed approach, consistently outperforming CLIP/MaskCLIP and methods using caption as supervision.

The AC went through the paper and the discussion and agreed with the reviewers' assessment. The article shows how we can learn a competitive open-vocabulary segmentation model without semantic labels is feasible, and this can be of interest to a large portion of the community, especially considering the new perspective under which the task has been addressed.

During the discussion, reviewers suggested several analyses and comparisons provided by the authors (e.g., MaskCLIP+, prompts, the impact of data/scalability, and centroids, among others). The authors should add these additional analyses and discussions in the final version of the manuscript and/or the appendix, as promised.